# Electrical Performance of ZTO Thin-Film Transistors and Inverters

**DOI:** 10.3390/mi16070751

**Published:** 2025-06-25

**Authors:** Jieyang Wang, Liang Guo, Xuefeng Chu, Fan Yang, Hansong Gao, Chao Wang, Yaodan Chi, Xiaotian Yang

**Affiliations:** 1Key Laboratory of Architectural Cold Climate Energy Management, Ministry of Education, Jilin Jianzhu University, Changchun 130118, China; jieyang202309@163.com (J.W.); stone2009@126.com (X.C.); ctpnxn@163.com (F.Y.); hsgao@jlju.edu.cn (H.G.); wangchao@jlju.edu.cn (C.W.); chiyaodan@jlju.edu.cn (Y.C.); 2School of Electrical and Computer Science, Jilin Jianzhu University, Changchun 130118, China; 3Department of Basic Science, Jilin Jianzhu University, Changchun 130118, China; 4Department of Chemistry, Jilin Normal University, Siping 136000, China

**Keywords:** annealing temperature, oxygen vacancy, ZTO, thin-film transistor, inverter

## Abstract

In this study, zinc–tin oxide (ZTO) thin films were prepared via radio-frequency magnetron sputtering to examine the influence of annealing temperature on the performance of thin-film transistors (TFTs) and their resistive-load inverters. The findings reveal that annealing modulates the concentration and spatial distribution of oxygen vacancies (V_O_), which directly affect carrier density and interface trap density, ultimately determining the electrical behavior of inverters. At the optimal annealing temperature of 600 °C, the V_O_ concentration was effectively moderated, resulting in a TFT with a mobility of 12.39 cm^2^ V^−1^ s^−1^, a threshold voltage of 6.13 V, an on/off current ratio of 1.09 × 10^8^, and a voltage gain of 11.77 in the corresponding inverter. However, when the V_O_ concentration deviated from this optimal range, whether in excess or deficiency, the gain was reduced and power consumption increased. This V_O_ engineering strategy enables the simultaneous optimization of both TFT and inverter performance without relying on rare elements, offering a promising pathway toward the development of low-cost, large-area, flexible, and transparent electronic devices.

## 1. Introduction

With the rapid advancement of display technologies and the integrated circuit industry, thin-film transistors (TFTs) have become increasingly important in applications such as displays, sensors and logic devices. Metal oxide semiconductors, including zinc oxide (ZnO) and amorphous indium gallium zinc oxide (α-IGZO), have emerged as promising alternatives to conventional hydrogenated amorphous silicon (α-Si:H) and polycrystalline silicon (poly-Si) because of their high electron mobility, compatibility with low-temperature processing, excellent stability and superior optical transmittance [1,2,3,4].

In recent years, extensive investigations into indium gallium zinc oxide (IGZO)-based thin-film transistors (TFTs) have predominantly focused on optimizing device architectures to enhance electrical performance and reduce power consumption. In pursuit of these objectives, scholars have implemented a range of strategies to improve the operational characteristics of IGZO TFTs [5,6]. For instance, full-swing inverters realized through top-gate biasing techniques have demonstrated significant improvements in switching behavior and noise margins [7]. Jin-Seong Park et al. further refined the electrical performance of resistive-load IGZO inverters, facilitating their practical deployment in display systems [8]. Additionally, the integration of ultrathin SnO_x_ capping layers for threshold voltage modulation has led to excellent voltage gain and swing, indicating strong potential for high-resolution display applications [9]. Moreover, optimization of the oxygen partial pressure ratio has been shown to improve the electrical characteristics of α-IGZO films, further enhancing inverter performance [10]. Despite these technical advancements, the scarcity and high cost of indium have prompted researchers to investigate alternative oxide semiconductors [11]. Among these, zinc–tin oxide (ZTO) has attracted growing attention as a promising indium-free, environmentally benign, and cost-effective material, owing to its favorable electronic properties and excellent thermal stability [12,13]. Previous studies have demonstrated that annealing temperature is a key parameter in tuning the electrical behavior of oxide-based TFTs, and that suitable thermal treatment allows ZTO devices to achieve performance comparable to their IGZO-based counterparts [8]. Annealing also plays a vital role in determining the structural and electronic properties of ZnSnO and ZTO films, as optimal conditions can reduce defect density and improve carrier mobility [14,15,16,17]. Nevertheless, systematic studies evaluating the inverter-level performance of ZTO-based devices remain relatively scarce, highlighting the need for further exploration in this domain.

Building upon the aforementioned challenges and previous findings, this study focuses on investigating the influence of the electrical characteristics of zinc–tin oxide (ZTO) thin-film transistors (TFTs) on inverter performance. Key parameters, including saturation mobility (μ_sat_), on/off current ratio (I_on_/I_off_), subthreshold swing (SS), and threshold voltage (Vth), are systematically analyzed to determine their impact on the electrical behavior of resistive-load inverters. ZTO TFTs were fabricated using radio-frequency magnetron sputtering, with the active layer deposited on silicon substrates coated with a SiO_2_ dielectric layer. Particular emphasis is placed on examining how annealing temperature modulates device electrical properties and affects inverter operation. To further assess the applicability of ZTO TFTs in logic circuits, resistive-load inverters were constructed and evaluated to explore the correlation between annealing conditions and overall device performance.

## 2. Experimental Process

In this paper, a 285 nm thick SiO_2_ is selected as the insulating layer, and the substrate is treated sequentially using acetone, anhydrous ethanol and deionized water. After cleaning, the substrate was blown dry using high-purity nitrogen, and photolithography was used to precisely define the key patterns of the device on the substrate surface. The active layers of the thin-film transistors were prepared by sequential deposition of active layers using the radio-frequency magnetron sputtering (RF sputtering) technique. The ZnSnO ceramic target used had a Zn to Sn ratio of 7:3, and the sputtering conditions were 90 W power and 8.5 min sputtering time. Before deposition, the pressure inside the chamber was maintained at 5 × 10^−5^ Torr; during sputtering, the working gas Ar to O_2_ ratio was 90%:10%. Subsequently, an annealing process was performed and secondary lithography was performed. Aluminum (Al) was evaporated using an electron beam evaporation (EB) apparatus to form a source (Source, S) and drain (Drain, D) with a film thickness of 50 nm. On the prepared TFT device, a 1 MΩ load resistor was connected to construct a resistive-load inverter. In this experiment, the number of devices tested for each annealing condition is as follows: approximately 20 devices for each condition (annealed at 400 °C, 500 °C, 600 °C, and 700 °C for one hour). One representative device from each condition was selected for measurement. The performance of the inverter is used for subsequent electrical characterization, and the device structure is shown in Figure 1.

## 3. Results and Discussion

### 3.1. Thin-Film Transmission Spectrum Analysis

As shown in Figure 2, the ZTO films prepared at different annealing temperatures exhibit an average transmittance of more than 90% in the visible range (400–800 nm), which is significantly higher than that of some transparent conductive films [18]. The annealing temperature has a significant effect on the optical transmittance of semiconductor films. The increase in transmittance can be attributed to the reduction in impurities in the films, which in turn reduces light scattering and enhances the optical transmittance of the films [19]. These results indicate that ZTO films have a wide range of potential applications as transparent conductive materials in transparent display devices.

### 3.2. Film AFM Characterization Analysis

In order to investigate the surface morphology and roughness of the ZTO thin films, atomic force microscopy (AFM) was used to characterize the samples. In this experiment, the roughness measurements and 3D models obtained from Atomic Force Microscopy (AFM) were processed using Asylum Research v16 software. The software was used to characterize the surface roughness and generate 3D topographical maps of the ZTO thin films. Table 1, on the other hand, lists the root-mean-square roughness (RMS) of the corresponding samples; Figure 3 shows the AFM 3D surface images of the ZTO films under different annealing temperature conditions. The experimental results show that as the annealing temperature is increased from 400 °C to 600 °C, the RMS roughness of the film surface decreases from 0.96 nm to 0.84 nm, indicating that the film surface tends to be flat. When the annealing temperature was further increased to 700 °C, the roughness slightly rebounded to 0.91 nm, which might be related to the reorganization of the crystal structure or the morphological changes induced by grain growth. A flatter surface helps to enhance the interface quality, which can effectively inhibit the generation of charge traps at the interface and reduce carrier scattering, thus increasing the carrier mobility of the channel and improving the electrical performance of the TFT [20]. The modulation of surface roughness is of great significance for realizing high-performance ZTO devices.

### 3.3. XPS Analysis of ZTO Thin Films

To further investigate the chemical composition of the ZTO films, we analyzed the chemical state of the oxygen element in the films using X-ray photoelectron spectroscopy (XPS). The O1s XPS spectra of ZTO films at different annealing temperatures are shown in Figure 4a–e. In order to eliminate the spectral line shifts due to charge effects, all binding energy data have been corrected by the C1s peak (284.8 eV). The O1s spectra have been decomposed into three Gaussian–Lorentzian-type peaks located at 530.15 eV, 531.50 eV, and 531.20 eV, respectively. The peaks at 530.15 eV (M-O) represent the oxygen bonded to the metal atoms (Zn, Sn), i.e., Zn-O, Sn-O bonds. The peak (V_O_) at 531.50 eV is associated with oxygen vacancies in the film, while the peak (M-OH) at 531.20 eV originates from oxygen-containing impurities, such as hydroxides [21]. The three subpeaks correspond to O_I_, O_II_ and O_III_, respectively. The relative area of each peak is shown in Figure 4a–e. O_I_ increases from 53.21% to 62.50% during the process of 400 °C–600 °C annealing. The M-O bonding increases and forms carrier transport channels in the oxide, and the high percentage of M-O helps in the transport of carriers, which in turn improves the carrier mobility of the device. O_II_ decreases from 30.30% to 27.54% because the temperature-elevated thermal oxidation process reduces the oxygen vacancies [22]. O_III_ decreases from 16.49% to 4.58% at 400–700 °C, and the hydroxyl groups are converted to oxides as the annealing temperature increases. When the annealing temperature was increased from 600 to 700 °C, the relative area of O_I_ decreased from 62.50% to 57.76%, and the relative area of the O_II_ peak increased again from 27.54% to 37.66%. This is because the weak bonding strengths of Zn-O, Sn-O and O can be easily diffused out from the ZTO film at high temperatures by a defect-assisted kick-out mechanism, the oxygen vacancies in the ZTO film act as shallow donors, and the reduction in the oxygen vacancies will lead to a decrease in the carrier concentration in the ZTO film [23,24]. The weak bonding strengths of the bonds, such as Zn-O, Sn-O, etc., are not strong enough to resist the thermal excitation, which leads to the fact that these bonds are more prone to fracture. The results show that the best electrical performance of the device can be inferred when the annealing temperature is 600 °C in agreement with the AFM measurements.

### 3.4. Electrical Performance of ZTO TFTs

In order to study the electrical performance of ZTO thin-film transistors (TFTs), this paper uses a semiconductor parameter tester to test the ZTO TFT, and the obtained device characteristic curves are shown in Figure 5. Figure 5a–e show the transfer characteristic and output characteristic curves of the ZTO TFT, respectively. The transfer characteristic curves are obtained by scanning at a source–drain voltage (V_DS_) of 40 V and a gate-to-source voltage (V_GS_) between −40 V and 40 V. The output characteristic curves demonstrate the relationship between the source-to-drain current (I_DS_) and the source-to-drain voltage (V_DS_) over the voltage range of 0 V to 40 V in steps of 5 V. The transfer characteristic curves are shown in Figure 5a–d.

As can be seen from Figure 5a–d, the source–drain currents of the samples at different annealing temperatures are positively correlated with the gate voltages and show a good saturation trend. This indicates that the ZTO TFTs prepared under different annealing temperature conditions have better gate voltage modulation performance. When the gate voltage is low, the source–drain current tends to be stable with the variation in source–drain voltage, while at higher gate voltage, the source–drain current rises rapidly with the increase in source–drain voltage until it reaches the saturation state. This further verifies that the device channel exhibits N-type operation mode. In the linear part of the transmission curve of Figure 5e, the relevant parameters shown in Table 2 were calculated by Equations (1)–(3).(1)$$\mu_{SAT}=\frac{2L}{WC_{i}}{\left(\frac{\partial \sqrt{I_{DS}}}{\partial V_{GS}}\right)}^{2}$$(2)$$I_{ON}/I_{OFF}=\frac{{\left(I_{DS}\right)}_{\max}}{{\left(I_{DS}\right)}_{\min}}$$(3)$$SS=\frac{dV_{GS}}{d(\log I_{DS})}$$
where C_i_ is the gate capacitance per unit area, W is a constant indicating the channel width, L is a constant indicating the channel length, V_GS_ and I_DS_ are indicated as the gate voltage and drain current, respectively, and (I_DS_)max and (I_DS_)min are indicated as the maximum source leakage current and minimum source leakage current.

Table 2 lists the electrical performance parameters of ZTO thin-film transistors (TFTs) prepared at different annealing temperatures. As the annealing temperature was increased from 400 °C to 600 °C, the carrier mobility increased from 9.12 cm^2^·V^−1^·s^−1^ to 12.39 cm^2^·V^−1^·s^−1^ and the threshold voltage decreased from 11.21 V to 6.13 V. This may be due to the densification of the films by the annealing process and the improvement of the electron transport properties. However, when the annealing temperature was further increased to 700 °C, the mobility decreased to 3.41 cm^2^·V^−1^·s^−1^ and the threshold voltage increased to 16.30 V. This change may be related to the reduced overlap of 4d^10^5s^0^ orbitals due to the increase in the oxygen content, as well as to the enhancement of ion scattering [17,25,26]. The concentration and distribution of oxygen vacancies play a crucial role in influencing these parameters. Moderate oxygen vacancies promoted the optimization of the ZTO thin films’ lattice structure, which increased carrier mobility and decreased threshold voltage. However, an excess of oxygen vacancies (as seen at 700 °C annealing) leads to increased carrier scattering, causing a decrease in mobility and an increase in threshold voltage. The subthreshold swing (SS) decreases and then increases with annealing temperature, reaching a minimum value of 0.78 V/decade at 600 °C. The switching ratio (I_on_/I_off_) reaches a maximum value of 1.09 × 10^8^ at 600 °C. The transistors prepared under annealing at 600 °C exhibit a significantly lower threshold voltage (6.13 V). This indicates that the lower number of oxygen vacancies at the interface enables the formation of conducting channels at lower gate voltages. The lower threshold voltage is conducive to reducing the power consumption of the device, which lays the foundation for the subsequent construction of low-power resistive-load inverters. In summary, the ZTO TFTs prepared under the 600 °C annealing treatment exhibit the best electrical performance.

### 3.5. Resistor-Load Inverter

As shown in Figure 6a, increasing the annealing temperature from 400 °C to 600 °C effectively suppresses the excess oxygen vacancies and increases the gain of the inverter from 8.02 to 11.77. When the temperature is further increased to 700 °C, the concentration of oxygen vacancies increases again, the interfacial quality is degraded, and the performance of the device decreases. Therefore, 600 °C is the optimal annealing temperature for the preparation of ZTO thin-film inverters. Figure 6b shows the static voltage transfer characteristic (VTC) of resistive-load inverter with ZTO TFT prepared at the optimum annealing temperature of 600 °C and connected in series with a 1 MΩ resistor. A total of five VTC curves are obtained under the conditions of input voltage V_in_ = 0–15 V and supply voltage V_DD_ = 10–30 V. All of them show clear inverter switching behavior, which verifies the good performance of 600 °C-annealed ZTO films in logic circuit applications. Figure 6c demonstrates the extraction method of the parameters V_OH_, V_OL_, V_IH_ and V_IL_ by taking the voltage transfer characteristic curve at V_DD_ = 30 V as an example. Figure 6d demonstrates the linear increase in V_gain_ and TW when V_DD_ goes from 10 V to 30 V, which is consistent with previous reports [27]. Table 3 lists the important switching parameters of the inverter at different V_DD_s, including voltage gain (V_gain_), high-level noise tolerance (N_MH_ = V_OH_ − V_IH_), low-level noise tolerance (N_ML_ = V_IL_ − V_OL_), and transition width (TW), etc. V_gain_ is defined as the maximum value of V_gain_ in the voltage transfer characteristic curve, i.e., V_gain_ = −(dV_out_/dV_in_), and TW is defined as TW = V_IH_ − V_IL_. At a V_DD_ of 10 V, the voltage gain is about 2.98, while at a V_DD_ of 30 V, the voltage gain increases to 11.77. It is important to note that a voltage gain of 2.98 is sufficient to drive the next level of components in the logic circuit [28]. The voltage swing of the inverter ([V_OH_ − V_OL_]/V_DD_ × 100%) gradually increases from 75.7% to 83.83% when V_DD_ is increased from 10 V to 30 V. This may be due to the fact that ZTO thin-film TFTs at optimum annealing temperature have higher on-state current (I_on_) and lower leakage current (I_off_) [29]. The wider voltage swing of the inverter improves the noise tolerance characteristics and makes the inverter operate more reliably in logic circuits [30].

## 4. Conclusions

In this paper, the effect of annealing temperature on the electrical characteristics of magnetron-sputtered Zn-Sn-O (ZTO) thin-film transistors (TFTs) and their resistive-load inverters is presented, and it is demonstrated that the concentration and distribution of the oxygen vacancies (V_O_) are important factors affecting the carrier transport, the density of interfacial traps, and the circuit gain. At the annealing temperature of 600 °C, the V_O_ is moderately suppressed; the film is dense and the interface is flat; the TFT mobility, threshold voltage, and switching ratio reach 12.39 cm^2^ V^−1^ s^−1^, 6.13 V, and 1.09 × 10^8^, respectively; and the resistive-load inverters constructed based on the film are shown to have the following characteristics. A voltage gain of 11.77 is achieved at a V_DD_ of 30 V, demonstrating its good ability to drive logic circuits. If the annealing temperature is further increased to 700 °C, the defect-assisted V_O_ concentration rises back up and the mobility plummets to 3.41 cm^2^ V^−1^ s^−1^, resulting in a significant degradation of the device performance, which verifies that too much or too little V_O_ affects the inverter performance. The above optimization of oxygen vacancies realizes the simultaneous optimization of TFT and inverter performance, which provides a feasible path and process paradigm for the industrialization of low-cost, flexible transparent electronic devices and low-power oxide logic circuits.

## Figures and Tables

**Figure 1 micromachines-16-00751-f001:**
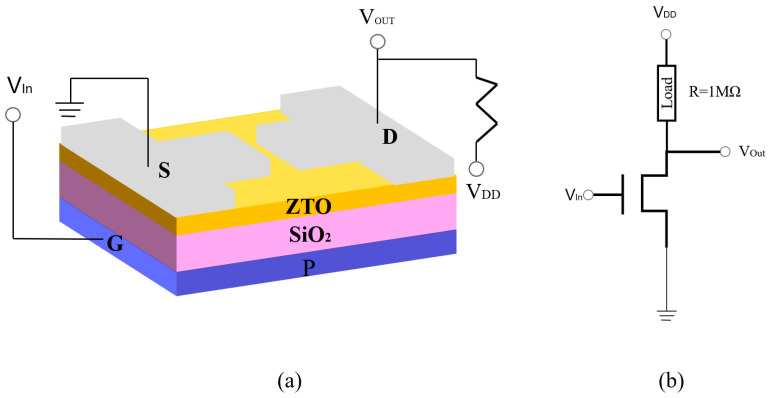
(**a**) Schematic diagram of the ZTO TFT-based resistive-load inverter; (**b**) circuit diagram of the resistive-load inverter.

**Figure 2 micromachines-16-00751-f002:**
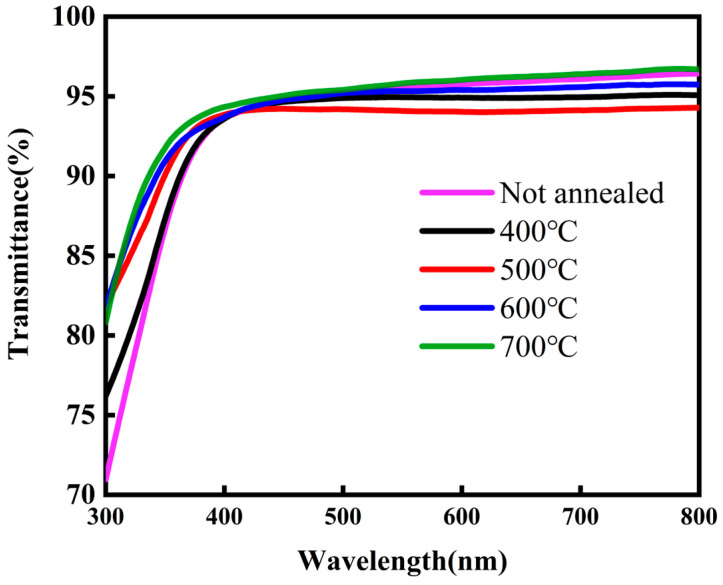
Transmission spectra of non-annealed and annealed ZTO films in the wavelength range of 300–800 nm.

**Figure 3 micromachines-16-00751-f003:**
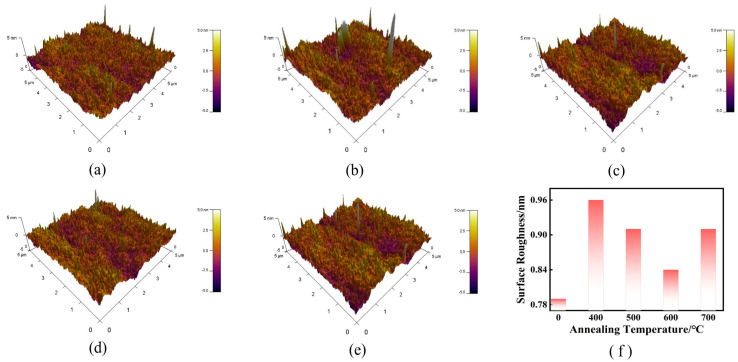
AFM test diagram and growth state of ZTO films annealed at different temperatures: (**a**) non-annealed, (**b**) 400 °C, (**c**) 500 °C, (**d**) 600 °C, (**e**) 700 °C, (**f**) surface roughness variation with annealing temperature.

**Figure 4 micromachines-16-00751-f004:**
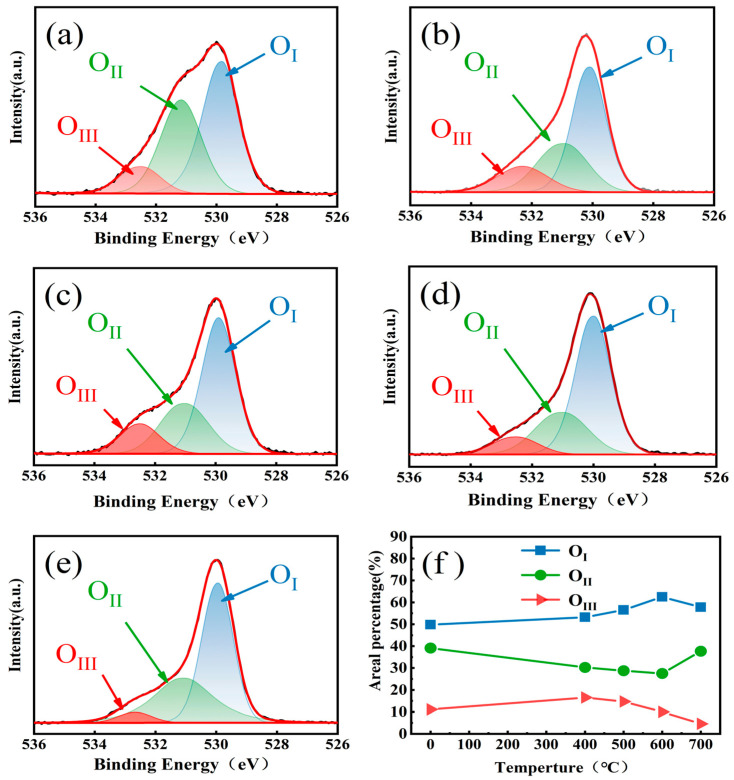
(**a**–**e**) O 1s XPS spectra and (**f**) contents of oxygen components of ZTO thin films with different annealing temperatures.

**Figure 5 micromachines-16-00751-f005:**
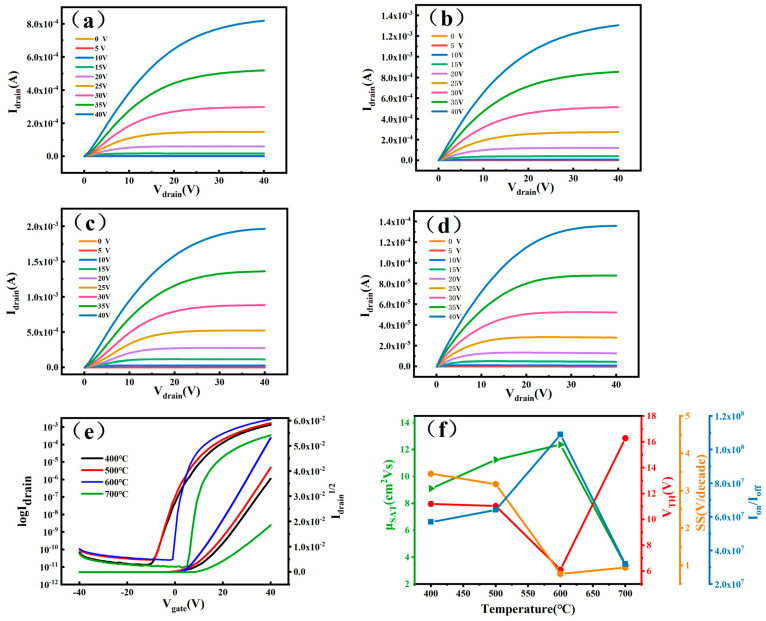
(**a**–**d**) Output characteristics of ZTO thin-film transistors after annealing at 400–700 °C. (**e**) Transfer characteristics curve at 400–700 °C. (**f**) The changes in mobility, subthreshold swing, threshold voltage, and current on/off ratio of the ZTO thin-film transistors as a function of annealing temperature.

**Figure 6 micromachines-16-00751-f006:**
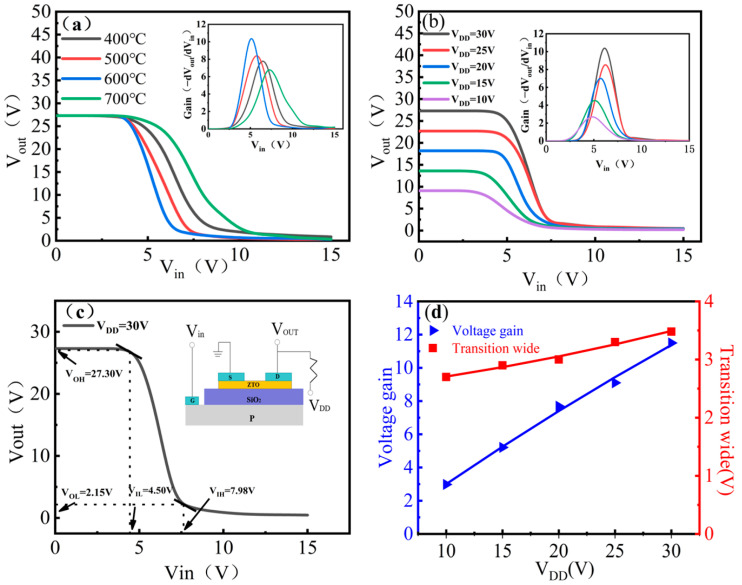
(**a**) Voltage transfer characteristics and gain of resistive-load ZTO TFT inverters under different annealing temperatures; (**b**) static voltage transfer characteristics of the inverter incorporating the 600 °C-annealed sample under various supply voltages (V_DD_). The inset shows the circuit diagram of the proposed inverter, which consists of a 600 °C-annealed ZTO TFT and a 1 MΩ resistor; (**c**) voltage transfer characteristic curve of the resistive-load inverter based on ZTO TFT at V_DD_ = 30 V; (**d**) transition width and voltage gain of the resistive-load inverter as functions of V_DD_.

**Table 1 micromachines-16-00751-t001:** Surface roughness of ZTO films not annealed and annealed at 400–700 °C.

Annealing Temperature/°C	Surface Roughness/nm
Not annealed	0.79
400	0.96
500	0.91
600	0.84
700	0.91

**Table 2 micromachines-16-00751-t002:** Electrical characteristics of ZTO TFTs at various annealing temperatures.

Annealing Temperature	Mobility (cm^2^ V^−1^ s^−1^)	Vth (V)	SS (Decade^−1^)	I_on_/I_off_
400	9.12	11.21	3.46	5.70 × 10^7^
500	11.26	11.06	3.18	6.42 × 10^7^
600	12.39	6.13	0.78	1.09 × 10^8^
700	3.41	16.30	0.95	3.20 × 10^7^

**Table 3 micromachines-16-00751-t003:** Representative device parameters of TFTs prepared from ZTO thin films annealed at 600 °C.

V_DD_ (V)	V_OH_ (V)	V_OL_ (V)	V_IL_ (V)	V_IH_ (V)	V_gain_ (V)	N_MH_ (V)	N_ML_ (V)	TW (V)
10	9.07	1.50	3.50	6.20	2.98	2.87	2.00	2.70
15	13.60	1.75	3.90	6.80	5.21	6.80	2.15	2.90
20	18.20	1.98	4.20	7.20	7.89	11.00	2.22	3.00
25	22.70	2.05	4.30	7.60	8.94	15.10	2.25	3.30
30	27.30	2.15	4.50	7.98	11.77	19.32	2.35	3.48

## Data Availability

The data that support the findings of this study are available from the corresponding authors upon reasonable request.

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
