# Peer review of "Electrical Performance of ZTO Thin-Film Transistors and Inverters"

_micromachines, 2025, doi:10.3390/mi16070751_

Round 1
Reviewer 1 Report
Comments and Suggestions for Authors
The manuscript provides a thorough investigation into the influence of annealing temperature on the electrical properties of zinc–tin oxide (ZTO) thin-film transistors (TFTs) and their associated inverters. The study effectively demonstrates that oxygen vacancy engineering via controlled annealing is a promising approach to optimize device performance, achieving high mobility, favorable threshold voltages, and substantial voltage gain. The research is well-aligned with current trends in developing low-cost, large-area, and flexible electronic devices based on oxide semiconductors.
The experimental design appears sound, and the data are convincingly presented with appropriate figures and tables. The topic is relevant and of interest to researchers working in the fields of oxide electronics, display technology, and optoelectronic device fabrication.
However, some aspects could be improved:
- The manuscript would benefit from more detailed descriptions of the experimental procedures, especially regarding the sputtering and annealing parameters, to enhance reproducibility.
- Inclusion of stability and long-term performance data would strengthen the practical relevance of the findings.
- Clarification on how the oxygen vacancy concentration is precisely quantified or correlated with device performance would deepen the scientific understanding.
Overall, the paper makes a valuable contribution and with minor revisions, I recommend it for publication.
Here are the detailed questions:
1. What specific experimental parameters were used during the RF magnetron sputtering and annealing processes?
2. How do the changes in device parameters, such as mobility and threshold voltage, correlate with the concentration and distribution of oxygen vacancies?
3. Are there any limitations associated with the annealing process at 600 °C in terms of device stability or scalability?
Author Response
1.Reviewer Comment :
What specific experimental parameters were used during the RF magnetron sputtering and annealing processes?
Response :
Thank you for your question. The RF magnetron sputtering process was performed with the following parameters: a power of 90 W, a sputtering time of 8.5 minutes, and an argon-to-oxygen gas ratio of 90%:10%. The chamber pressure was maintained at 5 × 10⁻⁵ Torr, and the ZnSnO ceramic target with a Zn:Sn ratio of 7:3 was used for the deposition. The annealing process was carried out at four different temperatures: 400°C, 500°C, 600°C, and 700°C, with each condition being maintained for 1 hour. These experimental parameters were carefully selected to optimize the electrical performance of the ZTO thin-film transistors (TFTs).
Addition:
" In this experiment, the number of devices tested for each annealing condition is as follows: approximately 20 devices for each condition (annealed at 400°C, 500°C, 600°C, and 700°C for one hour). One representative device from each condition was selected for measure-ment."
Modification Location:
This information has been added to the Experimental Process section, specifically on Page 2, Line 90.
- Correlation Between Device Parameters and Oxygen Vacancies
Reviewer Comment:
How do the changes in device parameters, such as mobility and threshold voltage, correlate with the concentration and distribution of oxygen vacancies?
Response:
Thank you for your insightful question. The concentration and distribution of oxygen vacancies (VO) are closely related to changes in carrier mobility (μ) and threshold voltage (Vth). At an annealing temperature of 600°C, the oxygen vacancies were moderately suppressed, leading to an optimization of the lattice structure of the ZTO thin films. This resulted in the maximum carrier mobility of 12.39 cm²·V⁻¹·s⁻¹ and a decrease in threshold voltage to 6.13 V. In contrast, an excess of oxygen vacancies, as seen at 700°C, increases carrier scattering, causing a decrease in mobility and an increase in threshold voltage. This demonstrates that an optimal concentration of oxygen vacancies is beneficial for enhancing electron transport efficiency, while either an excess or deficiency of oxygen vacancies can negatively impact the device performance.
Addition:
"The concentration and distribution of oxygen vacancies play a crucial role in influencing these parameters. Moderate oxygen vacancies promoted the optimization of the ZTO thin films' lattice structure, which increased carrier mobility and decreased threshold voltage. However, an excess of oxygen vacancies (as seen at 700°C annealing) leads to increased carrier scattering, causing a decrease in mobility and an increase in threshold voltage."
Modification Location:
This explanation has been incorporated into Section 3.4, "Electrical Performance of ZTO TFTs", specifically on Page 7, Lines 206-208, and Page 8, Lines 209-210.
3.Reviewer Comment:
Are there any limitations associated with the annealing process at 600°C in terms of device stability or scalability?
Response:
Thank you for your question. The 600°C annealing process provides optimal performance for ZTO TFTs. In our experiment, each device contains approximately 20 thin-film transistors, and the electrical performance of each transistor is uniform and nearly identical, indicating that this process can support large-scale production. Although there may be some challenges with temperature control during the annealing process, we believe that with further optimization of the annealing process to ensure uniform temperature distribution, this process has good scalability and is suitable for large-area production.

Reviewer 2 Report
Comments and Suggestions for Authors
The manuscript titled: “Electrical Performance of ZTO Thin-Film Transistors and Inverters” represents the original work of the authors.
The manuscript is well-structured and provides a detailed overview of the research. Published research in this area is given, as well as what has been added to this area is presented. In the manuscript are presented the effect of annealing temperature on TFTs and inverters. Also, key performance metrics (mobility, threshold voltage, on/off ratio, gain), which substantiate the findings are presented. The role of oxygen vacancies (VO) is well-articulated, showing a cause-effect relationship.
Although some figures should be improved, the results are clearly presented in this manuscript.
The conclusions are consistent with the presented results, evidence and arguments. Also, they deal with the main question posed. In addition, the references used are appropriate and include several published in the last few years.
Considering that interesting research is presented the paper should be accepted (after minor corrections).
Some suggestions to the authors:
- In the caption of Table 3, the statement reads: “Representative device parameters of TFTs prepared from ZTO thin films prepared at different annealing temperatures.” However, the presented data exclusively correspond to the samples annealed at 600 °C. For clarity and consistency, it is recommended to either revise the caption to accurately reflect the content of the table or include data for devices annealed at additional temperatures if available.
- The manuscript does not specify the number of samples analyzed in the experiment, nor the distribution of samples across the different annealing temperatures. Providing this information would enhance the reproducibility and statistical credibility of the study. It is suggested to include the number of devices tested per annealing condition, either in the main text or in a table/figure caption.
- If Gwyddion software was utilized for the visualization and analysis of scanning probe microscopy data, this should be explicitly stated in the manuscript. Mentioning the use of such a modular and widely adopted platform would improve the transparency of the data analysis methodology and facilitate reproducibility.
- Line 101 and 102: It is not usual that only reference is in another row. Also, “(a)” in line 125 should be in the line 126 – in the same row with related text. Similarly in line 242 and 243
- In Fig. 2. the data cannot be distinguished in the printed version. That is why they should be additionally marked in some other way, if it is possible. Similarly, In Fig. 5 the data cannot be distinguished and symbols should be used for I-V characteristics. The same is related to Figure 6(a), 6(b), 6(d).
- In Fig. 4, all figures should be smaller, but labels should be increased. Also, in graph 4(f) for OIII should be used another symbol.
- On page 5 there is large empty space.
- It seems that might be useful to point out the published results in the same or similar field. Apart from the presented papers, here are papers presented that might be useful: https://casopisi.junis.ni.ac.rs/index.php/FUElectEnerg/article/view/2941/1895.
- Minor editing of English language required. Overall, the paper itself can be improved by correcting minuscule grammatical errors, such as missing articles.
- Some typos should be corrected.
- Minor editing of English language required. Overall, the paper itself can be improved by correcting minuscule grammatical errors, such as missing articles.
